# Diagnostic Value of Imaging Methods in the Histological Four Grading of Hepatocellular Carcinoma

**DOI:** 10.3390/diagnostics10050321

**Published:** 2020-05-19

**Authors:** Feiqian Wang, Kazushi Numata, Masayuki Nakano, Mikiko Tanabe, Makoto Chuma, Hiromi Nihonmatsu, Akito Nozaki, Katsuaki Ogushi, Wen Luo, Litao Ruan, Masahiro Okada, Masako Otani, Yoshiaki Inayama, Shin Maeda

**Affiliations:** 1Gastroenterological Center, Yokohama City University Medical Center, Yokohama 232-0024, Japan; wangfeiqian@126.com (F.W.); chuma@yokohama-cu.ac.jp (M.C.); h.n.twopines@gmail.com (H.N.); akino@yokohama-cu.ac.jp (A.N.); k_ogushi@yokohama-cu.ac.jp (K.O.); 2Ultrasound Department, The First Affiliated Hospital of Xi’an Jiaotong University, Xi’an 710061, China; ruanlitao@163.com; 3Tokyo Central Pathology Laboratory, Hachioji 192-0024, Japan; masayukinakano23@gmail.com; 4Division of Diagnostic Pathology, Yokohama City University Medical Center, Kanagawa 232-0024, Japan; betana.m@gmail.com (M.T.); motani@yokohama-cu.ac.jp (M.O.); inayama@yokohama-cu.ac.jp (Y.I.); 5Department of Ultrasound, Xijing Hospital, Air Force Military Medical University, Xi’an 710032, China; lwdd1234@fmmu.edu.cn; 6Department of Radiology, Nihon University School of Medicine, Tokyo 173-8610, Japan; okada.masahiro@nihon-u.ac.jp; 7Division of Gastroenterology, Yokohama City University Graduate School of Medicine, Kanagawa 236-0004, Japan; smaeda@med.yokohama-cu.ac.jp

**Keywords:** hepatocellular carcinoma, contrast-enhanced ultrasound, histological grade, diagnosis

## Abstract

We attempted to establish an ultrasound (US) imaging-diagnostic system for histopathological grades of differentiation of hepatocellular carcinoma (HCC). We conducted a retrospective study of histopathologically confirmed 200 HCCs, classified as early (45 lesions), well- (31 lesions), moderately (68 lesions) or poorly differentiated (diff.) (56 lesions) HCCs. We performed grayscale US to estimate the presence/absence of halo and mosaic signs, Sonazoid contrast-enhanced US (CEUS) to determine vascularity (hypo/iso/hyper) of lesion in arterial and portal phase (PP), and echogenicity of lesion in post-vascular phase (PVP). All findings were of significance for the diagnosis of some (but not all) histological grades *(p* < 0.001–0.05). Combined findings with a relatively high diagnostic efficacy for early, poorly and moderately diff. HCC were a combination of absence of halo sign and isoechogenicity in PVP of CEUS (accuracy: 93.0%, AUC: 0.908), hypovascularity in PP (accuracy: 78.0%, area under the curve (AUC): 0.750), and a combination of isovascularity in PP and hypoechogenicity in PVP (accuracy: 75.0%, AUC: 0.739), respectively. On the other hand, neither any individual finding nor any combination of findings yielded an AUC of over 0.657 for the diagnosis of well-diff. HCC. Our study provides encouraging data on Sonazoid CEUS in the histological differential diagnosis of HCC, especially in early HCC, and the effectiveness of this imaging method should be further proved by prospective, large sample, multicenter studies.

## 1. Introduction

A histological classification method for hepatocellular carcinoma (HCC) can provide valuable information for determining the prognosis and selecting appropriate treatments, and perhaps even for allowing specific treatments to be selected for any given stage of HCC [1]. Earlier stages of HCC have been shown to be associated with a higher rate of surgical cure, a lower recurrence rate, as well as higher short-term and long-term survival rates as compared to the more advanced stages of HCC [2,3,4,5]. Specifically, early HCC is completely curable [6], while poorly differentiated (diff.) HCC is associated with the worst recurrence-free and overall survival rates [7]. Compared with patients with histologically diagnosed as well diff. HCCs, those who are diagnosed with moderately diff. HCCs tend to have a worse clinical stage of HCC [8] and a higher detectable vascular infiltration rate [9]. Therefore, it would be worthwhile to make an accurate diagnosis of the different histological grades of HCC. However, the currently available histological classification methods for HCC have limitations.

At present, a variety of histological classification methods are used, including the classification into early and advanced/progressive HCC [10,11], into well, moderately and poorly diff. HCC [12,13], and into grade I/II/III/IV HCC [14]. Some have even proposed classifications including very well diff. HCC [10] and “preinvasive” early HCC [11]. All of these histological classification methods have some drawbacks. Due to the heterogeneity or overlap of different histological classification methods, results of research related to the histological grading of HCC may be discrepant and inaccurate [15]. Furthermore, with the rapid development of diagnostic methods and treatments, more accurate and detailed classification methods than the traditional simple classification methods mentioned above are needed to obtain more histological information about HCC. Referring to the histological classification method into four grades proposed by Edmondson and Steiner [14], the histological classification methods of the International Consensus Group for Hepatocellular Neoplasia (ICGHN) [13] and International Working Party proposed in 1995 [12], we developed a novel histological classification method based on our clinical experience of more than 10 years in HCC diagnosis. We classified HCC into four grades—early, well diff., moderately diff., and poorly diff.—according to the staining patterns on sections stained with hematoxylin and eosin (HE), Victoria blue (VB) and silver stains, and immunostained for CD34 and cytokeratin (CK) 7.

In recent years, a number of reports on histological classification of HCC have been published. Histopathological examination of tumor biopsies or resected tumor specimens serve as the gold standard for determining the histological grades of tumors. However, it would certainly be desirable and worthwhile to explore non-invasive imaging approaches to obtain information on the histological grades of HCC at the earliest possible time and without performing an invasive biopsy [16]. The non-invasive imaging approaches can be classified into conventional imaging methods (e.g., ultrasound (US), CT, MRI), and the more novel contrast-enhanced imaging methods. Among the conventional methods, grayscale US is recommended by several guidelines and by consensus as an indispensable first-line screening method for HCC in high-risk populations (patients with hepatitis B, hepatitis C, liver cirrhosis) [17,18]; it is said to have overwhelming advantages, such as the following: relatively inexpensive, easy-to-perform, possibility of instant interpretation, reproducible in real time, and no radiation exposure [19]. However, there is little literature on the possibility of histological grading of HCC by grayscale US. Sonazoid contrast-enhanced ultrasound (CEUS) and gadolinium-ethoxybenzyl-diethylenetriamine pentaacetic acid magnetic resonance imaging (Gd-EOB-DTPA MRI) can provide rich information on the vascularity of the lesion and Kupffer cell function/hepatocyte function for the diagnosis of HCC. They have also been reported to show good performance for differential diagnosis of the histological grade, especially for the diagnosis of early-stage HCC [20,21]. Nevertheless, most previous studies have been limited to the evaluation of one or a few imaging methods or imaging patterns. In view of the advantages and disadvantages of various imaging examinations, we comprehensively utilized the findings of traditional grayscale US as well as those of novel imaging methods (CEUS and Gd-EOB-DTPA MRI) in the same patient population, in an attempt to obtain an accurate diagnosis of the histological classification of HCC.

In this paper, we present, in detail, our comprehensive analysis of the findings of various imaging examinations to identify patterns that could be useful for accurate diagnosis of the histological grades of HCC classified in accordance with the new four-grade histological classification system.

## 2. Materials and Methods

### 2.1. Patient Enrollment

A total of 200 consecutive liver biopsy specimens from 157 patients seen between January 2014 and August 2018 were analyzed in this study. All the lesions were newly discovered and as yet untreated, with histopathological confirmation of both the diagnosis of HCC and the histopathological grade of HCC. Clinical information (gender, age, etiology of cirrhosis, Child-Pugh grade), imaging data, and histology reports of the patients were collected by a retrospective review of the electronic medical records, radiology database, and pathology records of our hospital, respectively. Our retrospective study design was approved by the institutional review board of the Medical Center of Yokohama City University (Number B180200054) and in compliance with the principles of the Declaration of Helsinki; informed consent for the study was obtained from each of the participating patients. The criteria for exclusion from the present study were as follows:Child-Pugh grade C liver cirrhosis.Liver CEUS not performed within one month prior to the biopsy.No reliable or conclusive pathological diagnosis could be obtained because of insufficient biopsy specimens from the lesion, or a ‘‘no-hit’’ result for the target lesion during a percutaneous biopsy.Systemic chemotherapy or targeted treatment administered prior to the CEUS, which could potentially influence the findings on the CEUS images.

### 2.2. Grayscale US and Sonazoid CEUS Examination

Grayscale US and CEUS of the liver were performed at least once, within a month prior to the percutaneous US-guided liver biopsy. The equipment used was a LOGIQ E9 US system (GE Healthcare, Milwaukee, WI, USA) equipped with a native tissue harmonic grayscale imaging and CEUS function. The probes used were a convex probe with a frequency of 1–6 MHz and a micro-convex probe with a frequency of 2–5 MHz.

At first, grayscale US was performed. The size and position of the target hepatic lesion, the echogenicity of the lesion, the presence/absence of the internal mosaic sign and peripheral halo sign were observed and recorded. The diagnosis of “high”, “iso”, or “hypo” echogenicity was made on the basis of a more than 50% area of the lesion showing “high”, “iso”, or “hypo” echogenicity, respectively, as compared to the surrounding liver parenchyma. 

Then, Sonazoid CEUS was performed. A 0.2-mL dose of Sonazoid (Daiichi Sankyo, Tokyo, Japan) was injected into an antecubital vein at 0.2 mL/s via a 24-gauge cannula, followed by 2 mL of 5% glucose. CEUS images were acquired and evaluated during three contrast phases, namely, the arterial phase (AP) (10–50 s after initiation of injection), the portal phase (PP) (80–120 s after initiation of injection), and the post-vascular phase (PVP, also called the Kupffer phase) (10 min after initiation of injection). The vascularity of the lesions, as compared to that of the surrounding liver parenchyma, was classified as hyper-, iso-, or hypovascularity, when more than a 50% area of the lesion showed the above patterns, respectively. In general, the low mechanical index (MI) mode (0.21–0.28) (Figure 1d, Figure 2b,c, Figure 3c,d and Figure 4d) was used for the imaging, as it allows the entire liver to be observed in real time. Sometimes, we manually switched to low MI (0.18–0.28) harmonic imaging for the CEUS, which allows a more detailed evaluation of the tumor vessels and tumor staining, due to the high frame rate [22,23] (Figure 1b,c, Figure 3b, and Figure 4b,c).

In cases with hyperechoic lesions whose observation may be influenced by the background grayscale mode, we used high-MI (0.7–1.2) contrast imaging for more accurate evaluation of the presence/absence of a defective area during the PVP [23] (Figure 1e, Figure 2d and Figure 3e).

The US and CEUS examinations of all patients were performed together by two doctors who have been working in the field of liver US diagnosis for seven and 20 years, respectively. In view of the relative subjectivity of interpretation of the mosaic sign and halo sign in the US images, the presence/absence of these signs in each case was discussed and determined by two doctors. The CEUS images were also reviewed independently by the two doctors, a sonologist and a hepatologist, who had worked in the field of US diagnosis and liver disease for seven and 20 years, respectively. They were blinded to the final diagnosis, and the clinical and other radiological information about the patients. 

### 2.3. Gd-EOB-DTPA MRI

The majority of the enrolled patients (with 157/200 lesions, 78.5%) underwent Gd-EOB-DTPA MRI of the upper abdomen, within a month prior to the liver biopsy. A few patients did not undergo Gd-EOB-DTPA MRI, either because it was contraindicated (metallic implants in the body) and/or it was not necessary (HCC was clearly diagnosed by enhanced CT). The MRI was performed in a 1.5- or 3-T MR system (Signa HDx; GE Healthcare). Each patient received a rapid intravenous bolus injection of 0.25 mmol/kg body weight of gadoxetic acid (Gd-EOB-DTPA, Primovist^®^, Bayer Schering Pharma, Berlin, Germany) at the rate of 1 mL/s, followed by a 20-mL saline flush. Dynamic multiphasic (arterial-, portal-, and transitional-phase) imaging was performed using a fat-suppressed 3D T1WI gradient echogenicity sequence. The hepatobiliary phase (HBP) images were obtained 20 min after injection of contrast material. The signal intensity of each of the hepatic lesions, as compared to that of the surrounding liver parenchyma, was evaluated and the lesion was recorded as being a hypo-, iso- or hyperintensity when all or more than a 50% area of the lesion was visualized as a hypo-, iso- or hyperintensity, respectively. 

### 2.4. Diagnostic Criteria

After confirming the location of the HCC lesions by grayscale US/CEUS and Gd-EOB-DTPA MRI, a fine-needle aspiration biopsy was performed using a 21-gauge fine biopsy needle (SONOPSY; Hakko, Tokyo, Japan), under US/CEUS guidance. At least two samples from each lesion and one sample from the adjacent liver tissue (negative control) were obtained. All the tissue specimens were formalin-fixed, paraffin-embedded, and consecutively sectioned to the same thickness; the sections were stained first with HE, and then silver and VB stains, followed by immunostaining for CD34 and CK7. 

The HCCs were classified histologically according to the International Working Party criteria [12] and ICGHN [13], into early, well-diff., moderately diff., or poorly diff. HCC. We took well-, moderately and poorly diff. HCCs as a whole to be advanced HCCs. The features examined were the degree of cellular atypia, structural atypia, pattern of the reticular fibers, degree of neovascularization, degree of stromal invasion, and the ductular reaction (Table 1). The cell density, the nuclear size and morphology, the nuclear-cytoplasmic ratio, and alterations of the trabecular structure were evaluated in the sections stained with HE. (Figure 1g, Figure 2f, Figure 3f and Figure 4i). A decrease in the reticular fibers was evaluated using silver staining (Figure 1h, Figure 2g, Figure 3g and Figure 4j). The ICGHN arrived at the following consensus regarding the histological diagnostic criteria for early HCC [13]: it stated that the presence of tumor cell invasion into the intratumoral stroma (portal tract) [24] should be recognized as the most important histopathologic finding for the diagnosis of early HCC. The diagnosis of intratumoral stromal invasion requires VB staining [25] (Figure 1i). The ductular reaction, which refers to the staining of bile ducts in the periportal tract resulting from the change of bile metabolism in cases of HCC, was evaluated by immunostaining for CK7 [26] (Figure 1j). HCC sinusoids acquire the characteristics of capillary and precapillary blood vessels during de-differentiation from early to advanced HCC, and the tumor begins to reveal hypervascularity on angiography [27]. Neovascularization was examined by immunostaining for CD34, which is one of the most commonly used endothelial cell markers [27].

Immunohistochemical staining for CD34 (monoclone: QBEnd/10, mouse anti-Human; supplier: NOVO, Corp. Carpinteria, CA, USA; dilution: 1:100) and CK7 (monoclone: OV-TLJ 12/30, mouse anti-Human; supplier: DAKO, Corp., Carpinteria, CA, USA; dilution: 1:800) was performed using the LSAB kit, in accordance with the manufacturer’s instructions. It should be noted that CK7 can stain the following: (1) the cytoplasm of a few activated cancer cells (progenitor cells and intermediate hepatocytes) in some aggressive HCCs [28,29]; and (2) bile ducts around the portal tract (periportal tract) within the HCC lesion [26,29]. As bile-producing metabolic pathways in the malignant cells are abnormal in HCC, the expression of CK7 in the periportal bile ducts within the HCC lesion is decreased (early HCC) or absent (early HCC and advanced HCC) in HCC as compared to the findings in the adjacent, non-tumor parts of the liver. In line with the purpose of the study of histological grading of HCC, the observation of CK7 staining in the current study was confined to the bile ducts within the HCC lesion. CD34 staining was considered positive when positive staining was noted in the cytoplasm/membranes of vascular endothelial cells other than cancer cells. The CD34 staining pattern was classified as negative, focal positive or diffuse positive. “Negative” was defined as positive staining of only the blood vessels and bile ducts in the portal tracts and/or rare positive staining of the sinusoidal spaces (<10%) near the portal tracts. This pattern was observed in both the non-tumor areas and cases of early HCC. “Focal” was defined as positive staining of a small proportions (10–50%) of the sinusoidal spaces (Figure 1k). “Diffuse” was defined as positive staining of the majority of sinusoidal spaces (>50%) throughout the lesion area (Figure 2j, Figure 3j, and Figure 4m). 

The final histopathological diagnosis was made by the consensus of two expert pathologists with over 20 years of experience in liver pathology. In particular, one of the expert pathologists was a member of both the International Working Party and the ICGHN. Both the pathologists were blinded to the clinical, laboratorial, and imaging findings of the patients. Descriptive rather than numeric designations were used. Especially, when varying grades of differentiation were observed in different areas within a single tumor, we determined the diff. grade according to the worst diff. grade observed within the tumor [7].

### 2.5. Statistical Analysis

All the indicators examined in this research (baseline characteristics, imaging patterns, and final pathological diagnoses) were statistically analyzed. 

Age as a continuous variable (with normal distribution) was presented as the means ± standard deviation (SD), and the mean ages were compared among the four histological groups by one-way analysis of variance (ANOVA). Tumor size as a continuous variable (not with normal distribution, but with homogeneity of variance) was presented as the median values (interquartile range) and compared among the four histological groups by a non-parametric Kruskal–Wallis test. The remaining baseline data and imaging indicators were tested among the four histological groups by the chi-square test, chi square Linear-by-Linear association analysis, and Fisher’s exact test.

Two-tailed Fisher’s exact test and chi-square test were used for comparison of the frequencies of all the categorical variables to determine inter-group differences of the histological grade, where appropriate. The area under the curve (AUC) of the receiver operating characteristic curve was used to determine the diagnostic usefulness of indicators for each differentiation grade of HCC. The level of significance was set at *p* < 0.05. The cutoff values of continuous data (size of the lesion) were obtained from the receiver operating characteristic (ROC) curve analysis. The size value yielding the maximum sum of sensitivity and specificity is determined as the cutoff value. The above statistical analyses were performed using SPSS version 26.0 for Windows (SPSS Inc., Chicago, IL, USA). Overall missing or implausible data were excluded from the statistical analysis and extreme laboratory values were counted as missing. The sensitivity, specificity, and accuracy of each of the indicators were calculated manually.

## 3. Results

### 3.1. Baseline Characteristics

Table 2 shows the baseline characteristics of the study population with early, well-diff., moderately diff., and poorly diff. HCC. Of the total number of patients with HCC patients, 29, 111, and 60 patients had underlying HBV, HCV and other diseases (such as non-alcoholic steatohepatitis and primary biliary cirrhosis), respectively. There were no significant differences in the general characteristics (age, sex, etiology of HCC, Child-Pugh classification, location of the lesions) among the groups. The mean diameters (median (interquartile range)) of the early, well-diff., moderately diff. and poorly diff. HCCs were 14.00 (12.00–16.00), 17.00 (15.00–20.00), 17.50 (15.00–23.50), and 20.00 (12.00–26.50) mm, respectively. The size of lesion was defined by the maximum diameter measured in the US images (with reference to Gd-EOB-DTPA MRI and CEUS). Eight lesions showed a significant deviation of size (>100 mm in diameter) relative to that of the rest of the lesions and were beyond the measurement range of the US display screen. Considering the significant measurement error, they were regarded as invalid values and excluded from the statistical analysis of the sizes. The tumors differed significantly in size among the histological groups (*p* < 0.001, Table 3).

### 3.2. Comparison of the Grayscale US Patterns, and the CEUS and Gd-EOB-DTPA MRI Enhancement Patterns among the Four Histological Grades

When each of the imaging findings was analyzed independently (Table 3, *p*-value), the lesion size, presence/absence of the halo sign and mosaic sign in the grayscale US images, lesion intensity in each of the three phases of CEUS and in the HBP of Gd-EOB-DTPA MRI were significantly different in some histological grades. Specifically, the lesion size, presence/absence of the halo sign, and the lesion intensity in the AP and PVP allowed clear differentiation of early HCC from the other histological grades. Differences in the vascularity in the PP of CEUS were observed between poorly diff. HCC and all other histological grades of HCC. Between well-diff. and moderately diff. HCC, there were statistically significant differences in the positivity rate for the halo and mosaic signs, and findings in the PVP of CEUS. However, the findings on Gd-EOB-DTPA MRI allowed only differentiation between early HCC and poorly diff. HCC (*p* = 0.027). The echogenicity of the lesions on grayscale US was revealed to be of no significant value in the differential diagnosis of the histological grades of HCC (Figure 1a, Figure 2a, Figure 3a and Figure 4a).

### 3.3. Analysis of the Diagnostic Efficacy of Each of the Findings

Some of the findings in our study showed perfect-to-good efficacy for the diagnosis of early HCC (Table 4) and poorly diff. HCC (Table 4). The diagnostic efficacy for well-diff. and moderately diff. HCC (Table 4) was not very high (details of the calculation are not listed in this article). The best method to diagnose moderately diff. HCC was to determine the lesion vascularity in the PVP of CEUS, which was associated with an accuracy of 62.5% and an AUC of 0.697. Unfortunately, none of the imaging findings, either alone or in combination, showed an AUC of more than 0.657 or of any value for the diagnosis of well-diff. HCC (Table 4, data not completely shown).

The isoechoic pattern in PVP of CEUS showed the best efficacy for the diagnosis of early HCC (Figure 1d,e), with a sensitivity and specificity of 91.1% and 91.6%, respectively. The combination of isoechogenicity in PVP of CEUS and absence of the halo sign on grayscale US yielded the highest accuracy (96.1%) and a relatively high AUC (0.930) for the diagnosis of early HCC as compared to any imaging findings for any other histological grades.

A hypovascular pattern in PP of CEUS was the most valuable diagnostic indicator for poorly diff. HCC (Figure 4c). The sensitivity, specificity, and accuracy of the hypovascular pattern in PP of CEUS for the diagnosis of poorly diff. HCC were 71.7%, 80.3%, and 78.0%, respectively. However, an addition of the US findings in the AP of CEUS or PVP of CEUS or any other US findings for the diagnosis of poorly diff. HCC did not improve the efficacy, especially the accuracy and the AUC. Furthermore, as compared to other findings, the findings of the hypovascular pattern in PP of CEUS yielded the highest accuracy (88.7%) and AUC (0.780) for the diagnosis of poorly diff. HCC. 

For the diagnosis of moderately diff. HCC (Figure 3b–e), the combination of isovascular patterns in the PP and hypoechogenicity in PVP of CEUS was associated with acceptable diagnostic efficacy, with an AUC of 0.739 and an accuracy of 75.0%. As compared to other indicators for the diagnosis of well-diff. HCC (Figure 2a–c), the combination of hypervascularity in AP of CEUS and the isovascular pattern in PP of CEUS showed a relatively high diagnostic efficacy for well-diff. HCC, with an accuracy and an AUC of 62.0% and 0.657, respectively. 

The cutoff value of the lesion size obtained from the ROC curve analysis was 17.5 mm. Using this cutoff value, the lesions were divided into two groups: lesions measuring ≥18 mm and lesions measuring less than 18 mm in diameter. The lesion diameter differed significantly only between early HCC and other histological groups (Table 3, *p* < 0.001). Therefore, we conducted a subgroup analysis to determine the diagnostic value of each of the imaging findings for early HCC according to the size; the subgroup analysis showed that all the imaging findings that were found to be useful for the diagnosis of early HCC were still significant after adjustment for the lesion size (*p* < 0.05, data not shown). In particular, the AUC and accuracy of the most valuable diagnostic indicators (a combination of the halo sign and isovascularity in the PVP) were still high. The AUC for the subgroup with the smaller tumor size was 0.900, while that for the subgroup with the larger tumor size was 0.897. The diagnostic accuracy for the subgroup with the smaller tumor size was 90.0%, while that for the subgroup with the larger tumor size was 95.2% (Table 4).

## 4. Discussion

In our study, we found that using some imaging indicators alone or in combination yielded a high diagnostic efficacy for early HCC, as well as acceptable efficacy for poorly diff. and moderately diff. HCC. The diagnostic efficacy of the imaging indicators used alone or in combination proved to be much lower for well-diff. HCC as compared to that for the other three histological grades. Taken together, use of indicators derived from Sonazoid CEUS, such as the pattern of vascularity in the PP and echogenicity in the PVP, may be promising for diagnosis of the histological differentiation grade of HCC.

Grayscale US plays an important role in the diagnosis of HCC. Some of the latest clinical guidelines recommend that patients with hepatitis B, regardless of whether they have liver cirrhosis or not, must undergo regular US screening every six months [17,18]. A peripheral hypoechogenic “halo” of the lesion has been explained as being the radiological equivalent of a thin fibrous capsule in cases of advanced HCC [30]. It is believed to reflect the tendency towards expansive growth as large lesions begin to infiltrate the surrounding parenchyma as well as adjacent vessels and its branches [31]. The “mosaic” sign is positive in large tumors with a heterogeneous imaging appearance caused by multiple histological components (viable tumor with fat, necrosis and/or hemorrhagic components) [32], considered to be typical of advanced rather than small, early HCC. According to our study, both the “halo” and “mosaic” signs were useful to some extent for differential diagnosis of the histological differentiation grade of HCC. In particular, an absence of the “halo” sign was of significant usefulness for the diagnosis of early HCC, with a sensitivity as high as 95.65%. From the proportion of cases in the four histological groups showing positive “halo” (early: 4.4%; well-diff.: 41.9%; moderately diff.: 67.6%; poorly diff.: 57.1%) and “mosaic” (early: 2.2%; well-diff.: 16.2%; moderately diff.: 35.3%; poorly diff.: 35.7%) signs, the positive rate for “halo” signs revealed a sharply increasing trend with progression from early to moderately diff. HCC, while that for the “mosaic” sign seemed to show no such significant trend. Thus, our data suggested that the “halo” sign is of greater diagnostic value than the “mosaic” sign. On the other hand, our results showed that the echogenicity of a lesion was of no value for differentiating among the histological differentiation grades of HCC.

Hemodynamic changes taking place during hepatocarcinogenesis can be identified by CEUS. In general, the intranodular normal supply of hepatic arteries and portal veins decrease, while unpaired abnormal arteries and intranodular hepatic sinusoidal capillaries gradually increase in density with a worsening of the histological differentiation grade of HCC [33]. Almost all advanced HCCs are supplied entirely by the hepatic arterial system via abnormal unpaired arteries [34]. In our study, the findings in all three phases of Sonazoid CEUS were found to be of value in the differential diagnosis of the histological differentiation grade of HCC. Furthermore, in regard to specific analysis of correlation of the findings in each CEUS phase in relation to each grade, the efficacies were quite different. In regard to the findings in the AP, it has been reported that early and advanced HCCs classified on the basis of the histological grade usually show a hypovascular and hypervascular appearance, respectively [10,35]. In the PP and PVP, the washout time became shorter as the histological differentiation grade became poorer. Well-diff. HCC showed a more-delayed or no washout as compared to moderately and poorly diff. HCC [36,37,38]. 

The findings of our research are consistent with those of previous studies, but offer a more detailed differential diagnosis. From this study, we concluded that the most characteristic CEUS finding of early HCC is the isoechogenicity of the lesion in the PVP, suggestive of the absence of washout, in cases of early HCC. In contrast, washout began earliest (that is, in the PP rather than in the PVP) in cases of poorly diff. HCC, which can well explain the finding in our study that hypovascularity in the PP showed the best diagnostic efficacy for poorly diff. HCC. Notably, we found that the imaging findings in the PVP allow moderately diff. HCC to be differentiated from well-diff. HCC, whereas there were no significant differences in the findings in the AP or PP of CEUS between these two grades of differentiation. We also found that the findings of isovascularity in the PP and hypoechogenicity in the PVP can be of value in the diagnosis of moderately diff. HCC. Our results may suggest that the observer should pay more attention to the later phases of CEUS (PP and PVP) rather than to the earlier phase (AP). The strong ability of findings in the PVP of Sonazoid CEUS to allow different histological grades of HCC to be differentiated might be explained by the mechanism of progression of HCC: as HCC progresses, the Kupffer cells may become fewer or even disappear, resulting in an area clear of contrast material or a perfusion defect in Kupffer imaging [38]. This was further confirmed by the study of Imai, who used a kind of enhanced MRI to accurately determine the Kupffer cell numbers to predict the histological differentiation grade of HCC [39]. Nevertheless, we also found exceptions: for example, the presence of thrombosis in the portal vein or hepatic veins may contribute to vein flow reversal, exerting an influence on the hepatic circulatory dynamics and the diagnostic ability of CEUS [36]. The aforementioned explanations require to be verified through thorough and reliable data collection. 

Gd-EOB-DTPA has extracellular properties, which allow the blood supply of a lesion to be well-visualized. The “washout” of contrast materials from HCC lesions, and the different degrees of “washout” are supposed to be attributable to a gradual decrease of the portal blood supply with progression of the histological grade of HCC [40]. Moreover, Gd-EOB-DTPA is well-recognized for its hepatocyte-selective properties, which allow the functions of hepatocytes to be evaluated. During hepatocarcinogenesis, the dysfunctional hepatocytes show impaired uptake of contrast agents, thus allowing for visualization of hypointensity of HCC lesions as compared to the surrounding normal functional hepatocytes. Theoretically, different histological differentiation grades of HCC could be diagnosed based on different degrees of decrease in the blood supply and dysfunction of hepatocytes. A previous study reported a correlation between the relative enhancement ratio (the signal intensity of the tumor relative to the adjacent underlying liver) in the HBP of Gd-EOB-DTPA MRI and the histological differentiation grade of HCC [20]. Some studies have successfully combined findings in the HBP in Gd-EOB-DTPA MRI with some other parameters (quantitative analysis based on T1 mapping, hyperintensity on high-b-value diffusion weighted imaging, hypervascularity in the AP of Sonazoid CEUS) for accurate diagnosis or differential diagnosis of different histological differentiation grades of HCC [41,42,43]. These studies, with significant positive results, were more detailed than our study, as their measurements were quantitative (enhancement ratio) or more parameters were examined. We suppose that the limited information obtained from a qualitative observation of the HBP may be responsible for the lower significant level of the results of our study, as compared to the aforementioned previous studies. Therefore, if only basic qualitative analysis of the findings in the HBP of Gd-EOB-DTPA MRI is planned, we recommend avoiding the use of Gd-EOB-DTPA MRI for diagnosis of the histological differentiation grades of HCC.

In this study, the diagnostic ability of imaging findings was the lowest for well-diff. HCC (best accuracy and AUC are 62% and 0.657, respectively). First of all, it is easy to recognize that well-diff. HCC represents an intermediate stage of HCC progression (in terms of the gradual vascular changes of destruction and formation). Therefore, the characteristics of well-diff. HCC may overlap with those of early and moderately diff. HCC, so that no unique or obvious imaging indicators could be identified in cases of well-diff. HCC. In addition, the number of well-diff. HCC lesions in our study series was relatively small (31 lesions—less than half the number of moderately diff. lesions included in this study), which could reduce the statistical power to reliably identify its characteristics.

Previous studies have shown that some vascular appearances of HCC lesions on SonoVue CEUS (such as homogeneous hyperenhancement and washout time) may be affected by the tumor size [36]. We also found that the average sizes of moderately and poorly diff. HCCs were greater than those of less advanced HCCs. This phenomenon can be explained by the possibility that the well-developed fibrous septae in well-diff. HCCs pose an obstacle to the growth of the tumors [10]. However, the non-parametric Kruskal–Wallis test to compare tumor sizes between any two groups among the four histological groups showed significant differences in the tumor size only between early HCC and other more advanced HCCs. In other words, size does not seem to play an important role in the diagnosis of early HCC. Therefore, we performed a subgroup analysis using the most valuable indicators for the diagnosis of early HCC (combination of the halo sign and PVP) according to the tumor size. We found that the diagnostic efficacy of the findings was statistically perfect in both the subgroups classified by size, with an accuracy of more than 90% and an AUC higher than 89%. These findings may suggest that the tumor size had a negligible effect on the diagnostic efficacy of the imaging indicators and patterns identified by us. 

Our study had a few limitations. The first was the inadequacy of the biopsy specimens. It is generally acknowledged that histological diagnoses are far more difficult to make from biopsy specimens than from resected or autopsy specimens. Some studies have suggested that the potential problem of heterogeneous staining in biopsy specimens may reduce the diagnostic sensitivity of a biopsy as compared to resected specimens [44]. In view of this, when we recruited patients, a total of 30 patients who met all other criteria (such as performance of US and CEUS) were excluded because the biopsy did not hit the lesion, or too-little carcinoma tissue was obtained to allow a definitive histopathological diagnosis to be made. Secondly, our diagnosis of the histological differentiation grade of HCC was based on the prerequisite of a definitive diagnosis of HCC, which excluded some benign lesions such as regenerative nodules (RN). Most RNs show similar enhancement patterns (hypo, isovascularity in AP, PP, and isoechoic in PVP, respectively) to those of early HCC in Sonazoid CEUS [45]. However, RNs can be distinguished from early HCC by the central (92.3% of RN) and peripheral (97.6% of early HCC) vessel patterns in the AP [22]. In this case, if we want to make differential diagnosis of focal liver lesions (not just the histological grade of HCC), we need to pay more attention to AP.

## 5. Conclusions

To sum up, a detailed comparative analysis was made of the findings in commonly used non-invasive imaging examinations to determine the ability of individual findings or a combination of findings to allow diagnosis of the different histological differentiation grades of HCC. Most of the indicators produced positive results—especially a combination of an absence of halo signs in grayscale US and isoechoic in PVP in Sonazoid CEUS showed perfect results for diagnosing early HCC. We concluded from our study that early, poorly, and moderately diff. HCC according to our four classification method could be accurately or properly diagnosed independently or by a combination of certain patterns of image indicators (mainly Sonazoid CEUS). However, a proper diagnostic method for well-diff. HCC should be further explored. The promising results of our research will certainly be an indispensable step in towards a better application to clinical practice of our histological classification method.

## Figures and Tables

**Figure 1 diagnostics-10-00321-f001:**
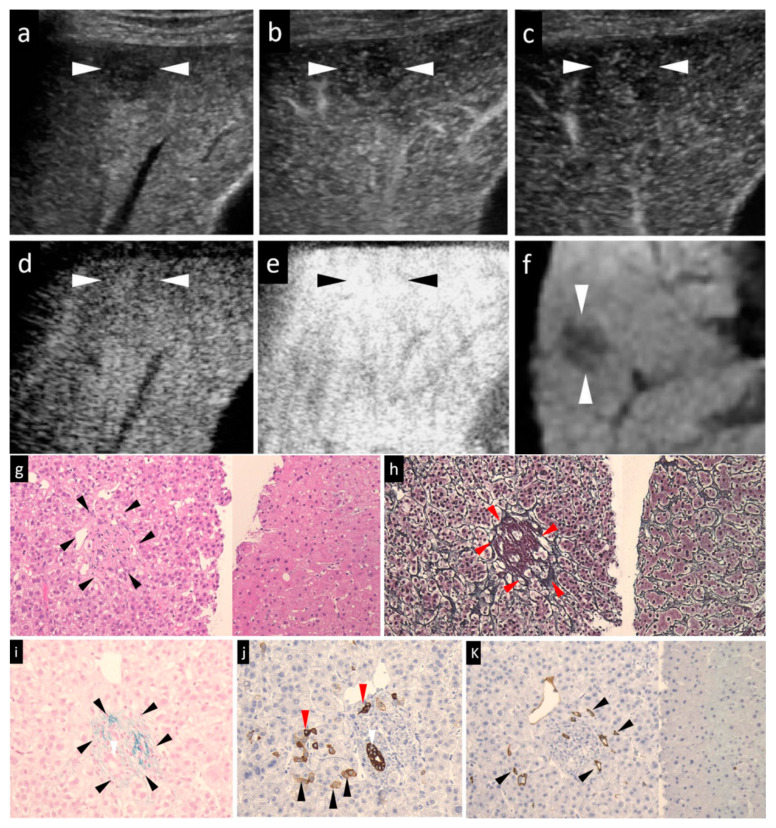
A case of early HCC (maximum diameter, 15 mm) in segment V with the typical imaging patterns and histological staining. The patient was a 65-year-old male. (**a**) Grayscale US showed a slightly hypoechoic lesion in segment V, near the surface of the liver. Neither the halo nor the mosaic sign was positive. (**b**,**c**) Arterial-phase Sonazoid CEUS image obtained using low-mechanical index (MI) harmonic imaging showed slight transient hypovascularity (**b**) and the lesion was seen as an isovascular lesion in the subsequent portal-phase image (**c**). (**d**) Post-vascular phase CEUS image obtained with low-MI contrast imaging showed an isoechoic tumor. (**e**) After the imaging mode was switched to high-MI contrast imaging, the post-vascular phase showed an isoechoic tumor. (**f**) The lesion was seen as a hypointensity in segment V in the hepatobiliary-phase image of Gadolinium ethoxybenzyl diethylenetriamine pentaacetic acid magnetic resonance imaging (Gd-EOB-DTPA MRI). (**g**) Hematoxylin–eosin (HE) staining. As compared to the non-tumor area (right side), early HCC (left side) showed the cancer area (arrowheads) has a slight cellular atypia, a mild increase in the nuclear-cytoplasmic ratio, and hypercellularity. The trabeculae were arranged clearly. (**h**) Silver staining of the tumor area (left side) and non-tumor area (right side). In early HCC, the reticular fibers were clear and evenly distributed, much resembling the findings in the non-tumor area. Red arrowheads indicated portal tracts. (**i**) Victoria blue staining of the tumor area showed elastic fibers surrounding the portal tract in blue (arrowheads). Positive stromal (portal tract) invasion, namely, cancer cells within the portal tract, were compatible with the diagnosis of early HCC. (**j**) Cytokeratin 7 staining of the tumor area. No ductular reaction could be seen in this figure. The progenitor cells were small cells with dense staining of the cytoplasm (red arrowheads). Black arrowheads indicated intermediate hepatocytes with faint staining of the cytoplasm. The white arrowheads in (**i**) and (**j**) indicated the presence of bile ducts within the portal tract. (**k**) CD34 staining. Focal expression of CD 34 (arrowheads) was seen in the cancer area (left side), which was compatible with the diagnosis of early HCC. Absence of CD34 expression in non-tumor area (right side) suggested the absence of any neovascularization in the non-tumor adjacent hepatic tissue. The arrowheads seen in images (**a**–**f**) indicated the margin of the lesion.

**Figure 2 diagnostics-10-00321-f002:**
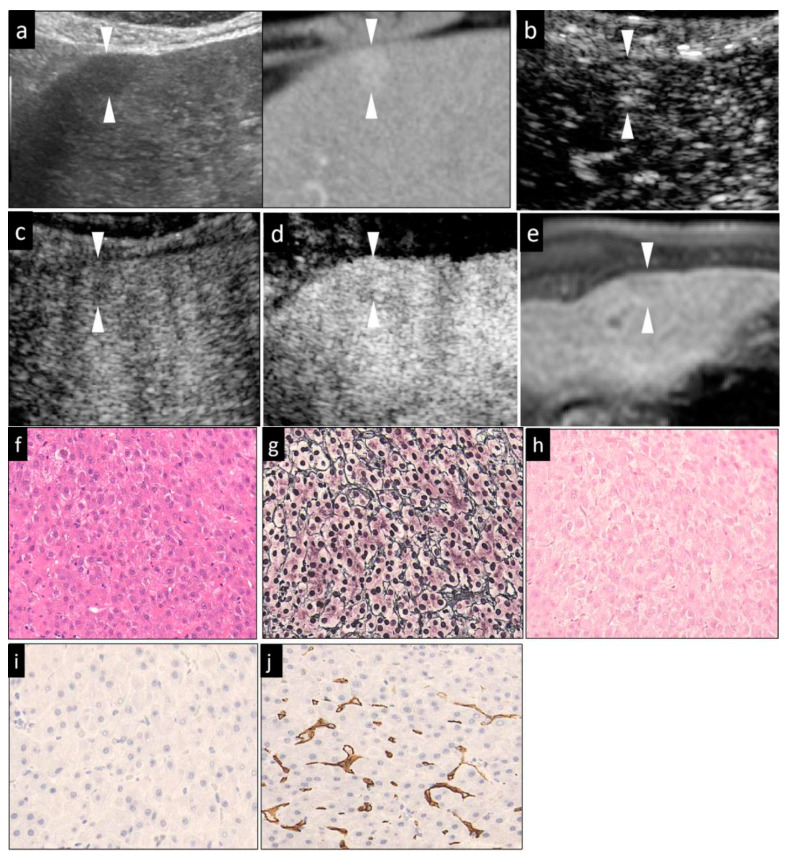
Mild hypervascular well-diff. HCC (maximum diameter, 13 mm) in segment III in a 72-year-old male patient with underlying chronic hepatitis C. (**a**) Fusion imaging combining grayscale US (left side) and the arterial phase of contrast-enhanced CT as reference (right side) on a single screen. The arterial phase contrast-enhanced CT image showed a small high-density area in segment III. The grayscale US image showed a homogeneous hypoechoic area with an unclear margin, and no halo around the lesion. (**b**,**c**) The arterial-phase Sonazoid CEUS image obtained using low-MI contrast imaging, (**b**) showed hypervascularity. The subsequent portal phase showed isovascularity (**c**). (**d**) After the imaging mode was switched to high-MI contrast imaging, the lesion was seen as an isoechoic lesion in the post-vascular phase. (**e**) The hepatobiliary phase of Gd-EOB-DTPA MRI showed a slight hypointense area in segment III, near the surface of the liver. (**f**) Hematoxylin–eosin (HE) staining showed slight cellular atypia, with an increased nuclear-cytoplasmic ratio and hypercellularity. The tumor cells showed a clear and thin trabecular cord pattern. The trabecular structure is distorted into a layer as thick as two cell layers. (**g**) Silver staining showed that the trabeculae are clearly arranged. (**h**) Victoria blue staining showing a negative stromal invasion (having cancer cells within portal tract) was compatible with the diagnosis of well-diff. HCC. (**i**) Cytokeratin 7 staining of the tumor area showed no ductular reaction. (**j**) CD34 staining. Expression of CD 34 was seen diffusely in the sinusoidal capillaries in the cancer area, which was compatible with the diagnosis of well-diff. HCC. The arrowheads seen in (**a**–**e**) indicated the margin of the lesion. The lesion was histopathologically diagnosed as a well-diff. HCC.

**Figure 3 diagnostics-10-00321-f003:**
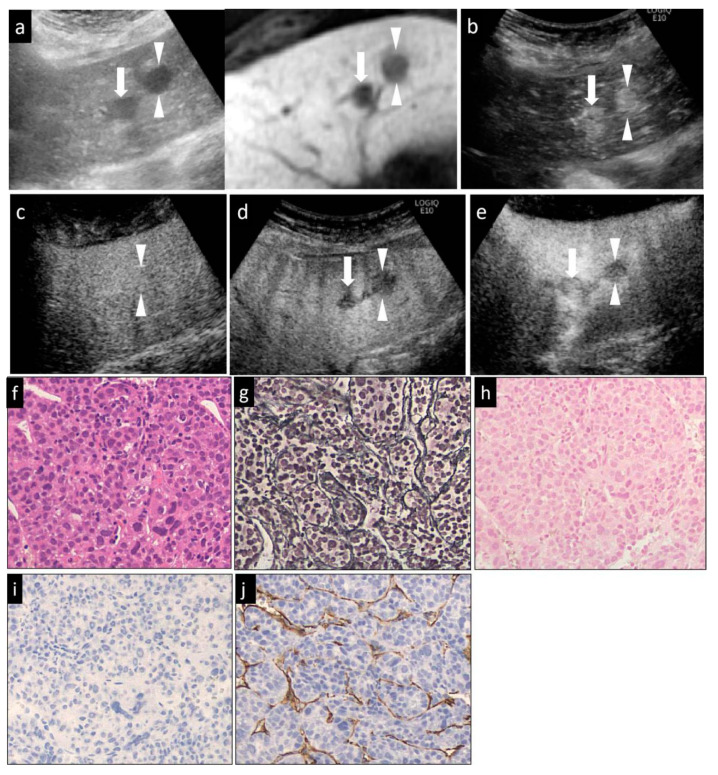
A hypervascular HCC lesion (maximum diameter, 16 mm) in segment III in a 58-year-old male patient with underlying chronic C hepatitis. (**a**) Fusion images combining grayscale US (left side) and the hepatobiliary-phase image of Gd-EOB-DTPA MRI as reference (right side) on a single screen. The hepatobiliary-phase image of Gd-EOB-DTPA MRI showed a significant hypointense area in segment III (arrowheads). The target lesion appeared as a well-defined hypoechoic lesion on the grayscale US image. The grayscale US image showed a homogeneous hyperechoic lesion without a surrounding halo. (**b**) Arterial-phase Sonazoid CEUS images obtained using low-MI harmonic imaging showed a significantly hypervascular area. (**c**) The lesion appeared isovascular in the portal-phase image, (**d**) and as a hypoechoic area in the post-vascular image (washout). (**e**) After the imaging mode was switched to high-MI contrast imaging, the lesion appeared as a clear defect in the post-vascular phase. (**f**) Hematoxylin–eosin (HE) staining showed obvious cancer cell and nuclear atypia, including hypercellularity, increased nuclear-cytoplasmic ratio and larger, deformed nuclei. (**g**) Silver staining showed that the reticulin formation was slightly unclear in some portions. (**h**) Victoria blue staining showed no stromal invasion of cancer cells. (**i**) Cytokeratin 7 staining showed no expression of the marker in the tumor areas, that was an absence of ductular reaction. (**j**) Diffuse expression of CD34 in the peripheral areas suggested increased neovascularization resulting from sinusoidal capillarization and a formation of sinusoidal vascular endothelium in HCC. This lesion was histopathologically diagnosed as a moderately diff. HCC. The arrowheads seen in (**a**–**e**) indicated the margin of the lesion. Another lesion was also seen in (**a**,**b**,**d**,**e**) (arrows).

**Figure 4 diagnostics-10-00321-f004:**
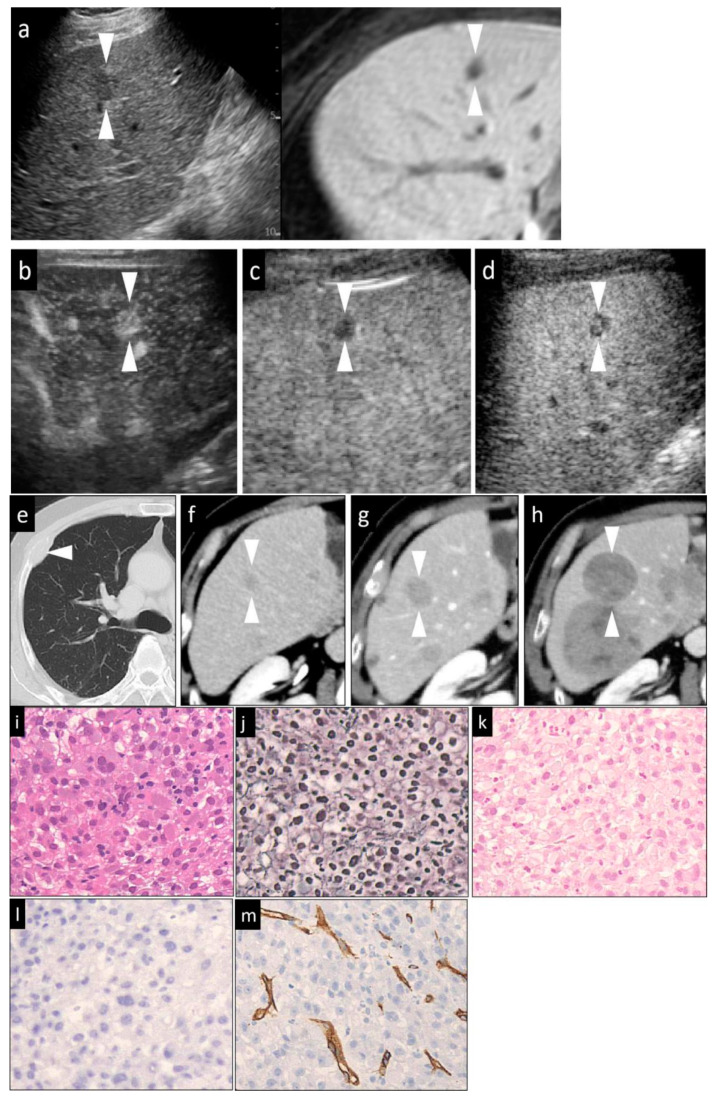
A hypervascular HCC lesion (maximum diameter, 16 mm) in segment V in an 82-year-old male patient with chronic C hepatitis. (**a**) Fusion images combining conventional US (left side) and hepatobiliary-phase image of Gd-EOB-DTPA MRI as reference (right side) on a single screen. The hepatobiliary-phase image of Gd-EOB-DTPA MRI showed a significantly hypointense area in segment V (arrowhead). The target lesion appeared as a well-defined, mild hypoechoic lesion, without a halo, on the conventional US image. (**b**) Arterial-phase Sonazoid CEUS image obtained using low-MI harmonic imaging showed hypervascularity of the lesion with a central non-enhancing area. (**c**) During the portal phase, the lesion appeared hypovascular (washout). (**d**) The post-vascular phase image showed a hypoechoic lesion. This patient received chemotherapy due to the presence of bone metastases (**e**). Contrast-enhanced CT image showing a small low-density area (arrowhead) prior to the chemotherapy (**f**). Contrast-enhanced CT image obtained at two months (**g**) and eight months (**h**) after the chemotherapy showed a marked increase in the size of the HCC lesion in a short period. (**i**) Hematoxylin–eosin (HE) staining. The tumor cells were arranged in solid sheets and the trabecular structure was lost. The nuclei were enlarged and have bizarre shapes. Multinucleated giant cancer cells could be seen. (**j**) Silver staining of a tumor area. The reticular fibers were almost completely lost. (**k**) Victoria blue staining showed negative staining, suggesting the absence of stromal invasion by the tumor cells. (**l**) Cytokeratin 7 staining of the tumor area. No staining was observed in the stroma within the cancer, indicating the absence of ductular reaction. (**m**) Positive immunoreactivity of the sinusoidal capillaries, indicative of diffuse CD34 expression. This lesion was histopathologically diagnosed as a poorly diff. HCC. The arrowheads seen in (**a**–**h**) (except for (**e**)) indicated the margin of the lesion.

**Table 1 diagnostics-10-00321-t001:** Our histological criteria for diagnosis of the differentiation grade of HCC ^1^.

Estimated Item			Cellular Atypia	Structural Atypia	Reticular Fibers	Neovascularization	Stromal Invasion	Ductular Reaction
Stain Method			HE	HE	Silver	CD34 ^2^	VB ^3^	CK7 ^4^
Histological differentiation grade	Early HCC		Slightly enlarged, non-spherical nuclei; Mild hyper-cellularity; slight increase of the N/C ratio	Normal or clear thin trabeculae	Clear and evenly distributed	Negative or focal	Presence	Decrease or absence
	Advanced HCC	Well-differentiated	Large, irregularly shaped of nucleus; hypercellularity; increased N/C ratio	Thick trabeculae	Clear and recognizable	Diffuse	Absence	Absence
		Moderately differentiated	Markedly enlarged, deformed nuclei; hypercellularity; increased N/C ration; multinucleated giant cells (occasionally seen)	Thick trabeculae or solid structures	Sometimes unclear and sparse	Diffuse	Absence	Absence
		Poorly differentiated	Large, bizarre-shaped nuclei; marked increase of the N/C ratio; multinucleated giant cells (frequently seen)	Disappeared trabecular and solid structure	Disappear	Diffuse	Absence	Absence

^1^ Abbreviations: HE: hematoxylin and eosin; VB: Victoria Blue; CK: Cytokeratin; N/C: nuclear cytoplasmic ratio; ^2^ Here, CD34-positive expression refers to positive staining of the sinusoidal capillaries. Focal and diffuse expressions of CD34 were defined as positive staining of only a proportion of the sinusoidal spaces (less than 50%) and staining of the majority of sinusoidal spaces (>50%) throughout the lesion, respectively; ^3^ Here, VB positive staining means staining of the portal tracts in the cancer lesion; thus, stromal invasion was defined as the presence of cancer cells within the portal tracts. If portal tract invasion was recognized within the HCC lesion, the grade of cancer cell atypia was irrelevant; ^4^ Here, as compared to the normal number of bile ducts around the portal tract (periportal bile ducts) in the non-tumor area of the liver, few or no periportal bile ducts (as determined by CK 7 immunohistochemistry) in the cancer lesion was classified as “decreased” or “absent” ductular reaction.

**Table 2 diagnostics-10-00321-t002:** Baseline characteristics of the enrolled patients and of the lesions ^1^.

Histological Grade	Early HCC	Well-Differentiated HCC	Moderately Differentiated HCC	Poorly Differentiated HCC	*p*-Value ^3^
**Patient characteristics**					
No. of patients	29	24	55	49	/
Age (mean ± SD, years)	70.33 ± 13.531	71.39 ± 8.036	70.07 ± 8.941	72.23 ± 10.248	0.493
Sex (Female/male)	10/35	6/25	19/49	11/45	0.673
Etiology of HCC (HCV/HBV/others ^2^)	24/5/16	19/4/8	38/9/21	30/11/15	0.743
Child-Pugh classification (Class A/B)	43/2	24/7	58/10	47/9	0.136
**Lesion characteristics**					
No. of lesions	45	31	68	56	/
Location (Left/Right hepatic lobe)	15/30	13/18	28/40	13/43	0.152
Size(s) (diameter(s)) (median (interquartile range), mm)	14.00 (12.00–16.00)	17.00 (15.00–20.00)	17.50 (15.00–23.50)	20.00 (12.00–26.50)	<0.001

^1^ Abbreviations: HCC: Hepatocellular carcinoma; HCV: hepatitis C virus; HBV: hepatitis B virus; ^2^ Others included etiologies, such as alcohol habit, non-HBV non-HCV, nonalcoholic steatohepatitis, and primary biliary cirrhosis. ^3^ Age showed normal distribution in each group and was tested by single factor analysis of variance (ANOVA). Tumor sizes were compared among different histological grades using the non-parametric Kruskal–Wallis test. The rest of the data were tested by the chi-square test or chi square Linear-by-Linear association analysis.

**Table 3 diagnostics-10-00321-t003:** Relationships between indicators and the histological grades of differentiation of HCC ^1^.

Indicators and Patterns	Characteristics	Group Comparison	*p*-Value
Size(s) (diameter(s)) of the lesions ^2^	/	Early and well differentiated	<0.001
		Early and moderately differentiated	<0.001
		Early and poorly differentiated	<0.001
Echogenicity on the grayscale US images	hyper/iso/hypo	total	0.286
Halo sign	positive/negative	Early (2/43) and well differentiated (13/18)	<0.001
		Early and moderately differentiated (46/22)	<0.001
		Early and poorly differentiated (32/24)	<0.001
		Well differentiated and moderately differentiated	0.016
Mosaic sign	positive/negative	Early (1/44) and moderately differentiated (24/44)	<0.001
		Early and poorly differentiated (20/36)	<0.001
Vascularity of AP in Sonazoid CEUS	hyper/iso/hypo	Early (13/6/26) and well differentiated (25/1/5)	<0.001
		Early and moderate (56/4/8)	<0.001
		Early and poorly (50/5/1)	<0.001
Vascularity in PP of Sonazoid CEUS	iso/hypo	Early (41/3) and moderate (49/17)	0.012
		Poorly (14/41) and early	<0.001
		Poorly and well (24/7)	<0.001
		Poorly and moderate	<0.001
Echo in PVP of Sonazoid CEUS	iso/hypo	Early (41/4) and well (12/19)	<0.001
		Early and moderate (1/67)	<0.001
		Early and poorly (0/56)	<0.001
		Poorly and well	<0.001
		Moderate and well	<0.001
Intense in HBP of Gd-EOB-DTPA MRI	low/iso or high	Early (40/0) and poorly (39/6)	0.027

^1^ Abbreviations: HCC: Hepatocellular carcinoma; CEUS: Contrast enhanced ultrasound; Gd-EOB-DTPA MRI: Gadolinium ethoxybenzyl diethylenetriamine pentaacetic acid magnetic resonance imaging; HBP: Hepatobiliary phase; AP: Arterial phase, PP: Portal phase; PVP: Post-vascular phase; ^2^ In order to obtain more information from the original data for the statistical analysis, the independent variable for comparison here is the specific value of the focus size (continuity variable), rather than the frequency of different size ranges according to the size cutoff value (classification variable). As the size of the lesion did not show normal distribution, the Kruskal–Wallis test in the rank sum test was used.

**Table 4 diagnostics-10-00321-t004:** Diagnostic efficacy for different histological differentiation grades of HCC ^1^.

Histological Differentiation Grade	Image Indicators and Patterns		Sensitivity (%)	Specificity (%)	Accuracy (%)	AUC (95% CI)
Early	Halo sign (Absence)		95.6	58.7	67.0	0.769 (0.700–0.837)
	Mosaic sign (Absence)		97.8	31.6	46.5	0.647 (0.567–0.727)
	Hypo in AP		57.8	91.0	83.5	0.744 (0.651–0.837)
	Iso in PP		89.1	43.5	54.0	0.666 (0.584–0.749)
	Iso in PVP		91.1	91.6	91.5	0.911 (0.856–0.967)
	Fully satisfying ”Iso in PP and PVP”		86.7	91.6	90.5	0.891 (0.828–0.954)
	Fully satisfying ”Iso in PVP and absence of halo sign”	total	86.7	94.8	93.0	0.908 (0.846–0.969)
		Size < 18 mm	87.2	91.8	90.0	0.900 (0.830–0.971)
		Size =>18 mm	83.3	96.1	95.2	0.897 (0.720–1.075)
Poorly differentiated	Halo sign (Presence)		56.6	57.1	57.0	0.569 (0.479–0.659)
	Hyper in AP		88.7	34.0	48.5	0.609 (0.524–0.690)
	Hypo in PP		71.7	80.3	78.0	0.750 (0.667–0.833)
	Hypo in PVP		100.0	36.7	53.3	0.683 (0.608–0.757)
	Hypo in both PP and PVP		69.8	81.6	78.5	0.747 (0.663–0.831)
Moderately differentiated ^2^	Fully satisfying “Iso in PP and Hypo in PVP”		70.6	77.3	75.0	0.739 (0.664–0.815)
Well-differentiated ^2^	Fully satisfying “Hypo in AP and iso in PP”		71.0	60.4	62.0	0.657 (0.554–0.759)

^1^ Abbreviations: HCC: Hepatocellular carcinoma; CEUS: Contrast-enhanced ultrasound; Gd-EOB-DTPA MRI: Gadolinium ethoxybenzyl diethylenetriamine pentaacetic acid magnetic resonance imaging; HBP: Hepatobiliary phase; AP: Arterial phase, PP: Portal phase; PVP: Post-vascular phase; AUC: Area under the curve; ^2^ After the individual and combined analysis of all the indicators, the indicators with the highest diagnostic efficacy for well- and moderately diff. HCC are listed in the table. The results for the other indicators and patterns with the lower diagnostic efficacy are not listed.

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
