# Peer review of "Diagnostic Value of Imaging Methods in the Histological Four Grading of Hepatocellular Carcinoma"

_diagnostics, 2020, doi:10.3390/diagnostics10050321_

Round 1

Reviewer 1 Report

This is another attempt to use ultrasound for differentiation of tumor grades of primary liver cancer, HCC. Ultrasound being suitable as a screening tool comparing to multiphase contrast enhanced CT or MRI, will have a profound impact if it can be used for tumor grading. Although this is a retrospective analysis with limitations such as biopsied histology evaluation as the authors pointed out, the results are useful to some readers even though not surprising. Detection of early HCC is of most clinical significance, and ultrasound is most suitable for this in comparison with other imaging modalities. For well- and moderately-differentiated HCCs, the significance of their identification was not discussed satisfactorily, for staging, patient selection (for treatment), prognosis? Please add this discussion briefly. Most important point for the authors to consider when they plan their future prospective studies for correlation: morphology-based evaluation including the classical histology grading is no longer sufficient despite of structure-function relationship. It is now the molecular classification, which illuminates the molecular underpins of HCC. There are plenty of recent publications on this, and the next study would incorporate sequencing data from the samples for molecular profiling instead of staining just for CD34 and CK7 for correlation with ultrasound findings. The other things to keep in mind is to associate ultrasound scans with circulating markers from liquid biopsy.

Author Response

Thank you very much for your valuable comments. We read carefully and reply point-to-point. Please see attachment. 

Reviewer 2 Report

  1. Since there are some discrepancies of interpretation grady scale US and CEUS between different operators, please describe how many operators perform the US imaging.
  2. As mentioned in your discussion section, the limitation of this study is all the histopathological samples are from biopsy specimen. However, I wonder there must be some patients who received surgical resection for their early stage HCC. If do so, how about the correlation between the histopathological findings and non-invasive imaging study?  
  3. In clinical practice, there are some considerable proportion of mixed HCC and CCC in a single hepatic nodule. What is the CEUS findings in such a case?

Author Response

We appreciate your insight comments. We made revision accordingly. Please see attachment.

Reviewer 3 Report

The paper is very interesting and the results are in line with other reports of the literature.

I suggest to do these small corrections:

Line 219: correct histological instead of hisotological

Line 317: In the caption of figure 1 g the author state that there is an arrowhead, but there is not any arrowhead. Please add it.

Author Response

We much thank the reviewer for his/her valuable comments. We modified our manuscript accordingly. Please see attachment.
